# A Mini-Review on Syngas Fermentation to Bio-Alcohols: Current Status and Challenges

Vishal Ahuja [1,2], Arvind Kumar Bhatt [3], Balasubramani Ravindran [4], Yung-Hun Yang [5,6] and Shashi Kant Bhatia [5,6,*]

1   University Institute of Biotechnology, Chandigarh University, Mohali 140413, Punjab, India
2   University Centre for Research & Development, Chandigarh University, Mohali 140413, Punjab, India
3   Department of Biotechnology, Himachal Pradesh University, Shimla 171005, Himachal Pradesh, India
4   Department of Environmental Energy and Engineering, Kyonggi University, Yeongtong-Gu, Suwon-si 16227, Republic of Korea
5   Department of Biological Engineering, College of Engineering, Konkuk University, Seoul 05029, Republic of Korea
6   Institute for Ubiquitous Information Technology and Applications, Konkuk University, Seoul 05029, Republic of Korea
*   Correspondence: shashibiotechhpu@gmail.com; Tel.: +82-2-3437-8360

**Abstract:** Biomass gasification produces syngas, mainly comprised of CO and $H_2$ along with $H_2S$, $CO_2$, $N_2$, and tar compounds. Inorganic carbon present in syngas as CO and $CO_2$ can be utilized for the production of several value-added chemicals including ethanol, higher alcohols, fuels, and hydrogen. However, chemical sequestration operates at a high temperature of 300–500 °C and pressure of 3–5 MPa in the presence of heavy metal catalysts. Catalyst regeneration and the maintenance of high temperature and pressure increased the cost of operation. Microorganisms like algae and bacteria including *Acetobacterium* and *Clostridium* also have the potential to sequester carbon from the gas phase. Research has emphasized the production of microbial metabolites with a high market value from syngas. However, scale-up and commercialization of technology have some obstacles like inefficient mass transfer, microbial contamination, inconsistency in syngas composition, and requirement for a clean-up process. The current review summarizes the recent advances in syngas production and utilization with special consideration of alcohol and energy-related products along with challenges for scale-up.

**Keywords:** biomass gasification; syngas utilization; microbial fermentation; alcohol

## 1. Introduction

Increased energy requirements and fuel consumption have raised environmental pollution and lowered air quality. During the inter-season period, burning agricultural residues (left out after harvesting) is a common practice that not only releases pollutant gases but also aerosols [1]. It has been found that biomass burning contributed around 10–70% of PM 2.5, and concerning health, PM 2.5 and PM 10 are among the most lethal pollutants [2]. Biomass burning contributes significantly, but it is not the main source of air pollution. Transportation, industries' operations including construction, power plants, and indoor emissions are also among the top air polluters [3,4]. Industrial operations alone contribute 23% $SO_2$, 15% CO, and 14% PM 2.5. On the other hand, transportation contributed 53% NOx, 18% CO, 17% PM 2.5, and 13% PM 10 while stubble burning shared only 14% CO and 12% PM 2.5 [2].

Reduced soil fertility and disturbed nutrient cycle are other major ill effects of air pollution [5–7]. Stubble burning raises the temperature of the soil and eliminates beneficial microorganisms. Reduction in microbial diversity affects the nutrient cycle [8,9] and the availability of nutrients for plants which ultimately deteriorates agricultural productivity.

Besides agricultural productivity, air pollution also leads to injury to leaves and grains, affects metabolism and enzyme activity, and promotes discoloration, chlorosis, and necrosis [2]. Air pollution exposure leads to skin and eye irritation, respiratory distress and neurological and cardiovascular disorders. Chronic exposure can lead to permanent health ailments like respiratory disorders, Chronic Obstructive Pulmonary Disease, and even cancer [2,10].

Air pollution not only leads to biodiversity loss and global temperature rise but also to the loss of GDP. An increase in global temperature by 3.2 °C can consume 18% of the worldwide GDP [11]. To control the financial loss and deterioration of the environment, the Paris agreement targets climate change and aims at limiting the global temperature rise by controlling GHG emissions well, and a standard temperature rise has been set to keep it within 2 °C (1.5 °C preferred) [12]. The Intergovernmental Panel on Climate Change suggested a preferred global warming level of 1.5 °C which can be accomplished only by lowering the global CO emission by at least 45% by 2030 and subsequently to net-zero emission by 2050. Transportation and industrial sectors have been identified as prime GHG emitters comprising 16% and 27% of the total emissions [11].

The severity of air pollution incidents has fueled the need for carbon-negative fuels, industrial processes, and the capture of already present $CO/CO_2$, which would aid in global decarbonisation. A transition toward renewable fuel/energy sources and utilization of renewable biomass from agricultural and carbon-rich waste as feed to carbon-negative fuels and other commodity chemicals also accelerate decarbonisation [13]. The carbon capture and storage (CCS) techniques are focused on the specific capturing of $CO_2$ from environmental air and industrial gaseous discharge using physical, chemical, or biological approaches and storing it till further use. The storage time relies on the method used for storage. The major hindrances to the selection of methods among various approaches employed for CCS are energy-intensiveness and associated cost [14].

Nowadays, another approach is getting much more attention than the conventional carbon capture approach: syngas production and fermentation. Syngas/synthesis gas is a mixture of $H_2/CO$ produced from the gasification of conventional fuels and hydrocarbons, and its reforming in the presence of oxygen, air, steam, or mixtures that react with the carbon source at elevated temperatures [15]. In this process, renewable biomass/residues from agricultural fields are getting special consideration for syngas production. Basically, syngas is an intermediate product as well as raw material for a variety of commodity chemicals and small organic compounds like organic acids, alcohols, hydrogen, jet fuels, alkenes, etc., via catalytic hydrogenation at higher temperatures and pressures [11,15] or biocatalysts [16] like *Clostridium kluyveri* [17], *Clostridium autoethanogenum* [18], and *Clostridium carboxidivorans* [19]. Most chemical catalysis referred to the use of metal-based catalysts like Ni [20], Fe-Ca oxides [21], Co, and Mo [22], but the conversion operates at a much higher temperature which makes the process costly and energy intensive. In comparison, microbial fermentation can work at mild conditions and efficiently transform it into commercial products like organic acid by *C. carboxidivorans* P7 [19], *Acetobacterium*, and *Desulfovibrio* dominated mixed culture [23], butanol by *C. carboxidivorans* [24], and ethanol by *Clostridium ljungdahlii* [25,26].

Microorganisms have shown potential for the utilization of CO as a carbon source and/or electron donor followed by their utilization via various metabolic pathways. For example, anaerobes follow the 'Wood-Ljungdahl pathway' [27] and sequester it into acetyl-CoA while, under an aerobic environment, carboxydotrophs used molecular oxygen for its oxidation to $CO_2$ followed by sequestration via pentose phosphate [28]. There is another group, chemolithoautotrophs, which oxidized CO by carbon monoxide dehydrogenase to $CO_2$ [29]. For all these pathways, acetyl-CoA is the junction point and its further utilisation depends upon the growth conditions prevailing. The pathways determining the end products are under the control of gene expression hence genetic modification might contribute to improve the fermentation rate and product formation.

Syngas fermentation represents an opportunity to transform waste gases into commercially viable products rather than leaving them to deteriorate the environment. The available literature also stated that inorganic as well as organic carbon can be converted into commodity products [30,31]. According to the Scopus database, a total of 439 articles (353 research articles and 86 review articles) have been published since 2014 on syngas production and its valorization using chemical and fermentation methods. The United State alone has published 98 articles, followed by China (86), Germany (56), South Korea (46), Spain (37), and the Netherlands (34) (Figure 1 Visualization network of countries involved in this area of research). A recently published article by Maki-Arvela discusses the use of various catalysts for syngas conversion into aviation fuel [32]. In another article, Liu et al. discussed different catalytic routes for ethanol production from syngas [33]. Very few articles have focused on the production of energy-related products from syngas via microbial fermentation. There is a review article focused on a broad aspect of biochemicals and biofuel production from syngas fermentation [30]. In the current review, recent trends in syngas fermentation for the production of ethanol and other energy-related products are summarized. Various parameters that affect the syngas fermentation process are discussed and techno-economic aspects are also evaluated for industrial as well as social acceptance of the process. The integration of processes with the currently available industrial process is emphasized to minimize pollution and make processes greener and more cost-effective (Figure 2).

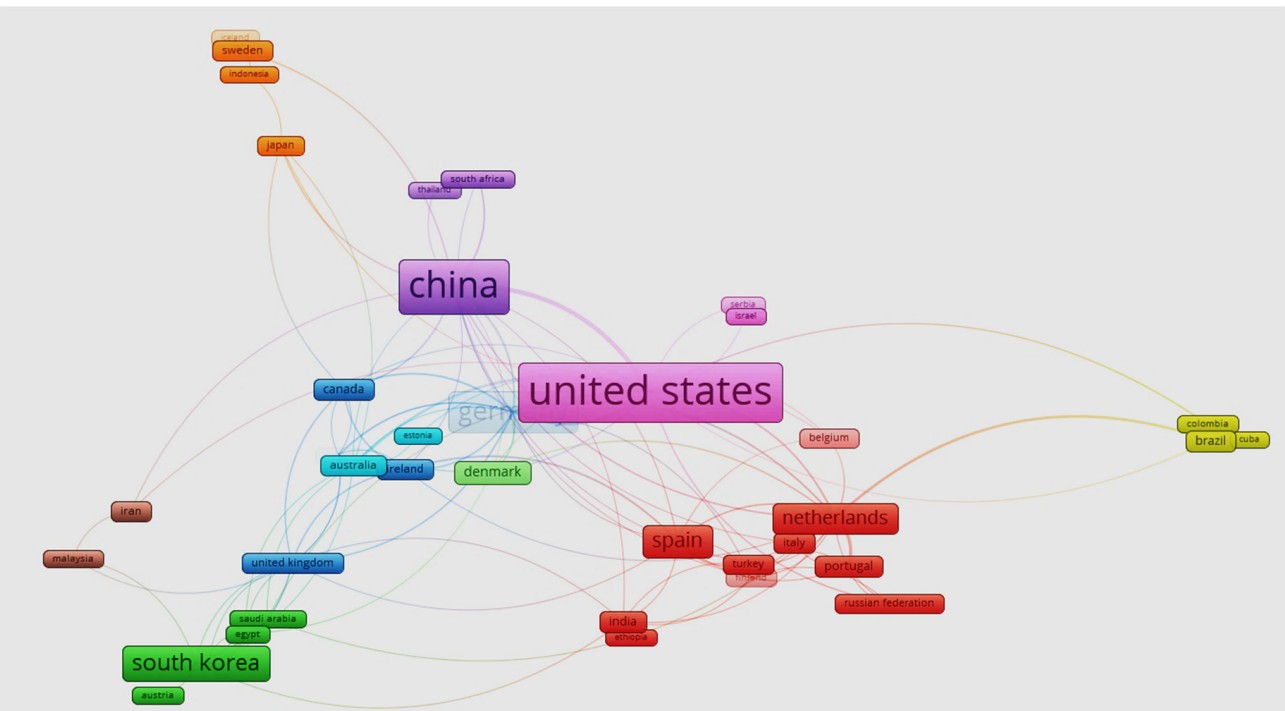

**Figure 1.** Co-occurrence mapping based on the publication number (minimum number of co-occurrences was fixed at 5). Map was generated using Vosviewer.

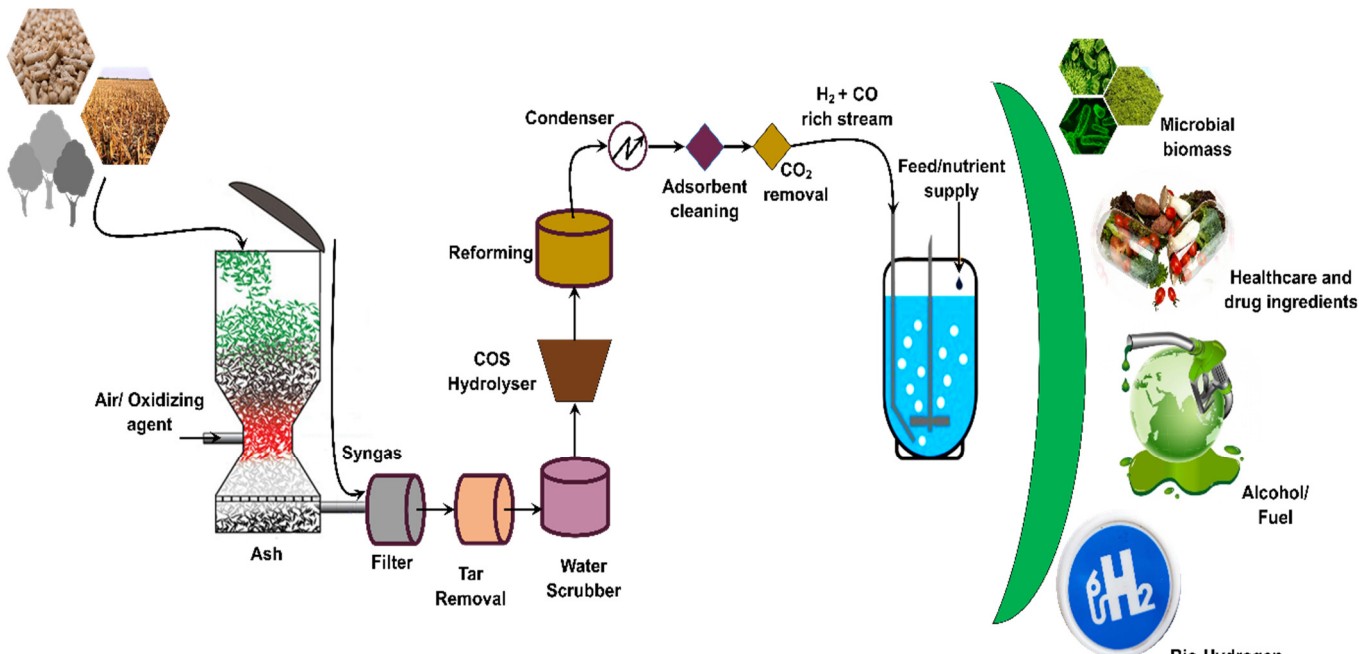

**Figure 2.** Process diagram showing syngas production technologies, pretreatment, and fermentation into various products.

## 2. Technologies for Syngas Production Pretreatment and Transformation

Syngas is mainly comprised of CO and $H_2$ as major components along with $CO_2$ and $H_2O$. It is a common product of biomass/fuel gasification with/without a catalyst. Usually, the process begins with a solid feedstock like coal or biological biomass including agricultural or microbial sources but may also include heavy oil. Before processing/gasification, biomass is dried and pulverized (if needed) for better results followed by gasification at a high temperature (800–1000 °C) and pressure of 1–20 bar [34]. The gasification process can be expressed in a reaction as follows:

$$Biomass + O_2 \rightarrow CO + H_2 + CO_2 + H_2O + CH_4$$

At the beginning of the gasification process, the solid carbon present in biomass/feed is partially oxidized in the presence of air/$O_2$/steam/or their mixture. The product is mostly represented by methane with a minor fraction of hydrocarbons like ethane and ethylene. The composition of gas may vary with the type of gasifiers, feedstock, and operational parameters, and product gas also carries some hydrocarbons and tar compounds like benzene, naphthalenes, and toluene. Besides ashes and char are also among the major by-products [34].

Xie and colleagues integrated catalytic pyrolysis and gasification for syngas production. Different types of nickel-based catalysts were evaluated along with pyrolysis temperature and gasification temperature. A temperature of 750 °C during pyrolysis and gasification was found good for higher syngas production with high-quality char. At 750 °C, char produced syngas with steam. A maximum syngas yield of 3.29 N m$^3$/kg feed was attained at 850 °C, but the catalyst got deactivated. At a higher temperature range, 'Ni' formed $Mg_{0.4}Ni_{0.6}O$, and the size of $Ni^0$ crystallite size increased [20].

Nourbakhsh and colleagues conducted Gibbs free energy minimization-based thermodynamic analysis for syngas utilization followed by experimental validation with ultra-rich methane/oxygen mixtures in an inert porous reactor via thermal-assisted partial oxidation. For the analysis, the equivalence ratio and the thermal load was considered the major factor. From the experimentation, the establishment of adiabatic conditions and heat loss abatement enhanced the syngas yield, and maximum syngas production (69.5%) was achieved

at an equivalence ratio of 2.5 with 8 kW energy investment [35]. Chen et al. used $CO_2$ as feed for direct conversion to syngas via the electrocatalytic process. From the process and scaleup point of view, the requirement of large current density, high cell voltage efficiency, and the design of efficient catalysts are critical points. Oxide-derived Cu nanowires with CuO heterostructures offered 90% $CO_2$ reduction with $H_2/CO$ ratios of 1, 2, and 3 along with 50% energy efficiency [36]. Hu et al. developed Fe-Ca oxides and employed them as oxygen carriers during syngas production to improve the hydrogen content via chemical looping gasification of rice straw as feed. The assessment was focused on the Fe/Ca ratio, reaction temperature, and oxygen carrier recyclability. Different combinations of Fe/Ca form calcium ferrites ($Ca_2Fe_2O_5$ and $CaFe_2O_4$) in which Fe and Ca were uniformly distributed. In comparison to control, hydrogen production was greatly enhanced due to steam chemical looping gasification as during looping steam re-oxidizes iron. The maximum yield of $H_2$ (23.07 mmol/g feed) was achieved with a Fe/Ca ratio of 1:1 which forms $Ca_2Fe_2O_5$ (oxygen carrier) at 800 °C. Rice husk is rich in silica which got accumulated on the catalyst during the process and destroys the Fe-Ca native structure after three cycles [21].

Gur and Canbaz conducted underground coal gasification (UCG) lab-scale experiment trials in the presence of hydrogen. The model was evaluated with a 2D-UCG computational fluid dynamics model considering syngas characteristics, formation of a cavity, reaction rate, and active zones. A two-stage gasification method was employed using lignite in a prism-shaped reactor in the presence of oxygen and steam (flow rate 5 $m^3$/h). The final product/syngas has 40 vol% hydrogens with a calorific value of 8 MJ/$Nm^3$ [37]. Dang et al. employed thermodynamics and biomass reaction kinetics for syngas production analysis. Data analysis identified the initial volatile composition from pyrolysis as a critical point for final product distribution while gasification temperature determines syngas composition and yield. Additionally, the moisture content in the feed, equivalence ratio, and steam-to-biomass ratio are also crucial parameters. Multivariable analysis-based optimization led to a maximum yield of 78.6 vol% at 900 °C from biomass feed with 30% moisture content, with an ER of 0.23 and a steam-to-biomass ratio of 0.21 [38].

Syngas is produced from the thermal process which not only has a high temperature but also carries particulate matter and tar compounds (depending upon the sources). Studies have suggested that syngas processing may become one of the necessary steps before fermentation as it might improve the microorganism's activities. Cleaning is a multistage process in which raw gases pass through water-trayed columns to eliminate fine char and ash particulate. Besides wet filters, cyclones, sieve filters, or candle filters are also used. In this stage, fine particulates, chlorides, ammonia salts, and traces of $H_2S$ are removed [39,40]. After scrubbing, gas is passed through sulfided, activated carbon beds which removed a major fraction (>90%) of the mercury and heavy metal from syngas [41,42]. Syngas usually contains acid gases including $H_2S$, carbonyl sulfide 'COS', and $CO_2$ near the ambient temperature of 37.7 °C. Acid gases are removed by passing the gas from regenerative solvents like methanol, dimethyl ether, methyl diethanolamine, rectisol, etc., in an absorber column [43].

Wilson et al. compared twenty zeolites, three activated carbons, two activated aluminas, and two silica gels for $CO_2$ and CO separation at 30 °C. As the concentration of CO and $CO_2$ changes with production methods and process conditions, different types of adsorbents may be needed for syngas cleaning. Among twenty-seven samples, high-density silica gel and H-Y-type zeolites were most effective in $CO_2$ removal from syngas. This was due to the shape of their isotherms which offered higher $CO_2$ adsorption capacities contributed by $Na^+$ exchange in zeolites at low pressures [44]. Monir and colleagues compared the effect of syngas processing and the removal of tarring compounds on microorganisms during a syngas fermentation trial for ethanol production. Among two syngas streams used, one was used as it is and and one was processed by passing through methanol and acetone (to remove tar compounds) followed by a cotton filter (to remove particulate matter). Syngas impurities in the untreated stream have an adverse effect on cell growth. Until the second day, bacterial growth was rapid followed by a stationary phase. Till the

11th day, growth appeared to be steady, then bacteria were not even visible. In the case of treated syngas, a lag phase was observed in bacterial growth during which bioethanol was produced. Comparative analysis revealed that cell growth in the treated stream was 500 times higher in comparison to the untreated one [45]. Liakakou et al. also processed the syngas produced from beech wood and lignin-rich feedback (received from biorefinery). Both the syngas contained tar compounds and sulfur-containing compounds. The gas was cleaned by a series of cleaning chambers equipped with host gas filters, activated carbon, OLGA, and TNO cleaning systems along with a CoMoO catalyst-based desulfurization unit. After cleaning, the gas streams were used for microbial fermentation for acetate and ethanol production [22].

Cali and colleagues integrated an air-blown gasifier of 5 MWth with a wet scrubber for syngas production and in-line cleaning. The whole unit is combined with a wastewater management system. Gasifier converted stone pine and eucalyptus feed to syngas with a conversion efficiency of 79–80%. In the cleaning unit system, tar and dust particles were removed via wet cleaning followed by water scrubbing. From here on, the outlet passed to waste management and regeneration systems like oil skimmers and activated carbon-based absorbent systems to regenerate the water followed by its reutilisation [46].

Chatrattanawet et al. simulated the syngas production followed by its cleaning using AspenPlus™ software [47]. It was found that syngas yield increased with the rise in temperature and achieved an optimum yield at 750 °C and a molar flow rate of ~149 kmol/h. The air gasification unit offered a higher syngas molar flow rate (A/B: 1.309) than steam-air gasification. Gasification was followed by absorption-based cleaning to capture $CO_2$ and $H_2S$ using monoethanolamine. The cleaning process was operated at 40 bars of column pressure with 10 trays. As a result, $H_2S$ content reduced to >0.1 mg/m$^3$ at monoethanolamine with all kinds of gasification systems comprised of steam, air, and steam-air having molar flow rates of 325, 450, and 465 kmol/h, respectively [47]. Frilund and colleagues developed a hybrid activated carbon and ZnO-based adsorbent for syngas cleaning. It was reported that the adsorbent successfully removed almost all syngas contaminants of biomass origin and the results were better than the wet scrubbing system [48].

*2.1. Chemical-Based Methods for Syngas Conversion into Bio-Alcohol*

Syngas ($H_2$/CO) can act as ideal feedstock for the production of numerous chemicals used as fuel/fuel additives and polymers. Chemical catalysis is one of the preferred methods used for syngas conversion [49]. However, this relies upon heterogenous or heavy metal catalysts for the purpose. Among various chemicals, ethanol and butanol are among the major products that can be employed as fuel/fuel additives and act as carriers for hydrogen in fuel cells.

Spivey and Egbebi developed a process for ethanol production from biomass-derived syngas under thermodynamically practicable operating conditions, i.e., 350 °C at 30 bar. Under the given conditions, selective conversion was very low (<10%), and ethanol production was further lowered if methane was also allowed as a product. Maximum ethanol production was achieved with an Rh-based catalyst [50]. Metal-based hybrid catalyst (Cu-ZnO-metal loaded ZSM-5) was evaluated for the conversion of syngas to gasoline-ranged hydrocarbons at 543 K in near-critical n-hexane. Cu-ZnO catalyzed the hydrogenation of CO to methanol and finally to hydrocarbons over the metal-loaded ZSM-5 via dimethyl ether. Among the selected metals, i.e., Pd, Co, Fe, and Cu, hydrocarbon yields of 59% and 64% were obtained with Pd/ZSM-5 and Cu/ZSM-5, respectively, at 5% metal loading. A further increase in Cu loading in ZSM-5 lowered ether and improved hydrocarbon yield. In addition, Cu/ZSM-5 hybrid catalyst was not deactivated for 30 h of the reaction [51].

Kang and colleagues employed triple tandem catalysis for syngas-to-ethanol conversion. The catalyst was comprised of potassium-modified ZnO–ZrO$_2^+$ modified zeolite mordenite + Pt–Sn/SiC. These three catalysts work in a single reactor but in a sequential manner as K$^+$–ZnO–ZrO$_2$ catalysed the hydrogenation of syngas to methanol, mordenite fraction carboxylate methanol to acetic acid, and Pt–Sn/SiC further hydrogenate it to ethanol. The

maximum syngas-to-ethanol conversion was reported with 90% selectivity [49]. Kaithal et al. reported the conducive effect of alcohol for CO hydrogenation to methanol under the catalysis of molecular manganese complex [Mn(CO)$_2$Br[HN(C$_2$H$_4$PiPr$_2$)$_2$]] ([HN(C$_2$H$_4$PiPr$_2$)$_2$] = MACHO-iPr). The complex attained the turnover number of 4023 with a turnover frequency of 857/h in the presence of EtOH/toluene as solvent at 150 °C, 5/50 bar (pCO/H$_2$), in 8–12 h. The reaction attained a selectivity of >99% under optimum conditions without accumulation of formate ester [52]. Table 1 summarizes some of the major products produced from syngas via chemical catalysis.

**Table 1.** Commodity chemicals' synthesis from syngas.

| Product | Chemical Catalysts | Raw Material | Operating Condition | Output/Yield | References |
|---|---|---|---|---|---|
| Ethanol | Potassium-modified ZnO–ZrO$_2$ + modified zeolite mordenite + Pt–Sn/SiC | Syngas | Fixed-bed flow reactor syngas flow rate: 25 mL/min; pressure: 5.0 MP; temp: 550 K | 90% selectivity for ethanol production | [49] |
| Ethanol | Rh–Mn impregnated on Zr-UiO MOFs | Syngas | - | 322.0 g/(kg.h) | [53] |
| Ethanol | CuZnAlOOH catalyst | Syngas (H$_2$/CO ratio 2), | Pressure: 4.5 MPa; temp: 553 K; GHSV 1/4 450 mL/g.h; mixing rate 700 rpm | 33.7% selectivity | [54] |
| Dimethyl ether | Cu/ZnO/Al$_2$O$_3$/ ferrierite catalyst | Syngas | - | 91–97% conversion | [55] |
| Ethanol | Carbon layer (H–MOR–DA@C) coated mordenite zeolite and Pt–Sn/CNT | Methanol + syngas | - | 60% selectivity with 98% conversion | [56] |
| Alcohol | ZnO/ZrO2 modified CuCoAl catalysts | Syngas | Temp: 30 °C; pH 8.0–8.5 | 42.6 wt% selectivity (C2 + OH/ROH 83.7%) | [57] |
| Ethanol | CuZnAl slurry catalysts | Syngas (H$_2$ + CO ration: 2), | Temp: 250 °C; pressure: 4.0 MPa | 45% selectivity (50% ethanol of total alcohol) | [58] |
| Higher alcohol | Co-Co$_2$C catalysts + Fe | CO + H$_2$ | - | 57.1% selectivity | [59] |
| Linear Aldehydes and Alcohols from Alkenes | Dual Rh/Ru catalyst | Formic acid + Syngas | Temp: 90 °C; period: 24 h | 88% | [60] |
| C2 Oxigenates | RhMn nanoparticles were fixed within siliceous MFI zeolite crystals | CO + H$_2$ (ratio 2:1) | Catalyst: 0.5 g; temp: 320 °C; pressure: 3 MPa; H$_2$/CO: 2; flow rate: 30 mL/min | Oxygenate selectivity: 40.3% CO conversion: 42.4% C2-oxygenate selectivity: 88.3% | [61] |
| C3 alcohol | K-Co-MoSx catalyst | CO + H$_2$ (1:1) | Temp: 360 °C; pressure: 8.7 MPa; GHSV: 4500 mL/g·h | C3+ alcohol selectivity 31.0%; product yield 9.2%; CO conversion: 29.8% | [62] |
| Higher alcohols | Carbin nanofibers supported K-promoted CoMo(5 wt%)-based catalysts | Syngas | Temp: 723 K; ambient pressure | Higher selectivity 22% | [63] |
| C2 alcohol | Cu$_{0.25}$Co$_{0.75}$ catalysts | CO + H$_2$ | Temp: 300 °C, time 4 h in H$_2$ atmosphere Pressure 2 MPa | Alcohol: 147.65 g/kg.h C$_{2+}$OH: 81.5% | [64] |
| Alcohol | La − 51.9% + Co-22% + Na-0.05% + Si-8.1% | Syngas | GHSV 450/h; Gas flow rate 1 L/h | 80% conversion | [65] |

It is clear from the above examples that chemical catalysis offered very high selectivity and tenability for syngas conversion to alcohols and other products, but like other processes, it also operates at a very high-temperature range, from 350–600 °C, and pressure. In addition, the cost of catalysts is also very high in the case of Rh, Pd, Pt, etc., which need regeneration after regular intervals. The process also employed solvents which may be hazardous. Overall, technical feasibility, process cost, and environmental suitability make it more challenging to find some eco-friendly alternatives. In order to make the conversion process sustainable and cost-effective over chemical catalysis, biological seques-

trations have been adopted. This approach emphasized the utilization of major components of syngas, i.e., CO, $H_2$, and in some cases COx and NOx as well in regular metabolic pathways of microorganisms via assimilation into various value-added chemicals and microbial biomass.

However, biomass gasification of carbonaceous fuels including agricultural biomass, petrochemicals, etc., in the presence of controlled oxidants like oxygen itself or air also generates SOx, NOx, and methane along with CO and $CO_2$. The exact composition of syngas may vary with the fuel/feedstock source and operating conditions [66].

### 2.2. Biological Sequestration and Syngas Fermentation

Microorganisms import component gases (CO, $H_2$, and $CO_2$) from syngas and metabolic enzymes oxidize/reduce these and finally use them for regular metabolism. It has been found that in anaerobes the Wood-Ljungdahl pathway is critical for the sequestration of CO as well as $CO_2$ while aerobic microorganisms utilize two different mechanisms. chemolithoautotrophs oxidize CO to $CO_2$ in the presence of carbon monoxide dehydrogenase while another group of microorganisms, i.e., carboxydotrophs, utilizes molecular oxygen and oxidizes CO. The resultant $CO_2$ is sequestered to the regular metabolic pathway via the pentose pathway and leads to glycolysis and the TCA cycle.

The Wood-Ljungdahl pathway can be classified into the carbonyl phase and methyl phase pathway [27]. Both the branches are connected at the acetyl-CoA junction, and from here, the successive reactions determine the fate of the molecules like the synthesis of the biomass or some other value-added chemicals including alcohol, acetates, acids, etc. (Figure 3; Table 2).

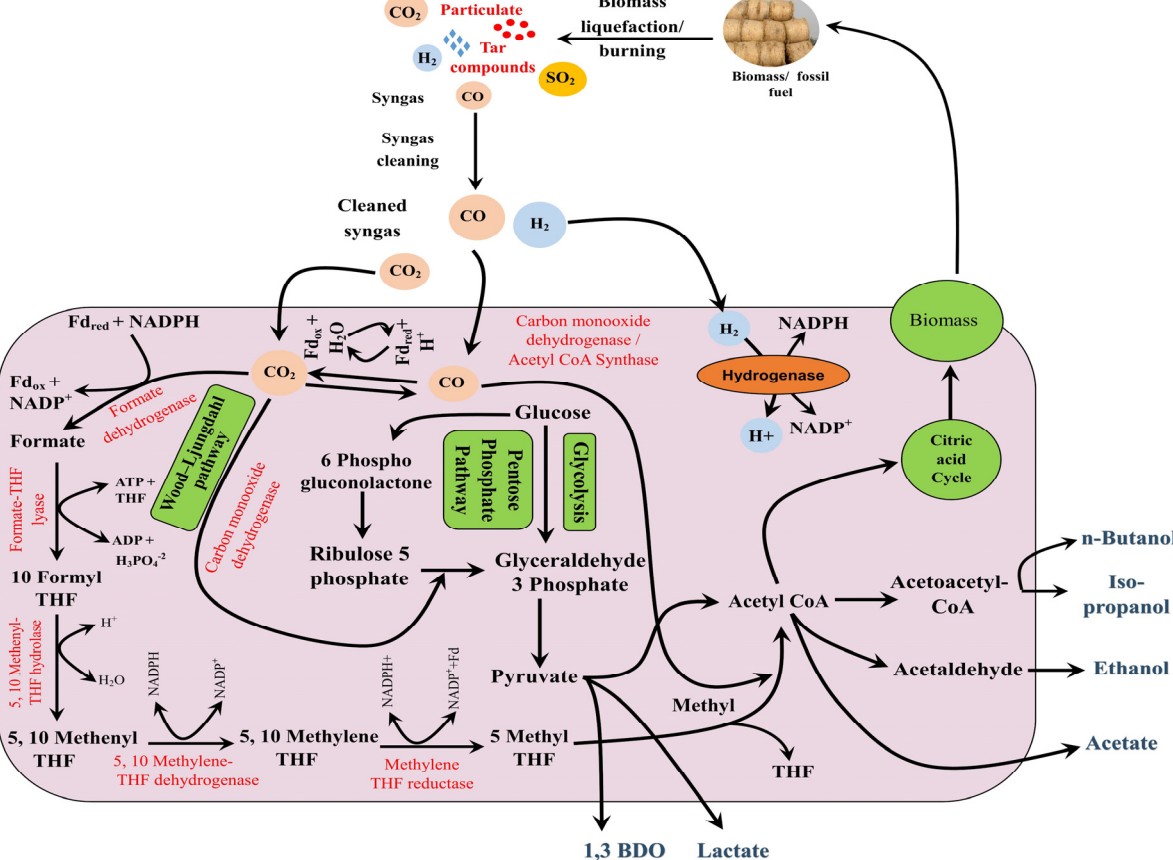

**Figure 3.** Sequestration of syngas and production of microbial metabolites.

Anaerobic bacteria follow the 'Wood-Ljungdahl pathway' for the utilization of CO and $CO_2$ sequestration. For syngas fermentation sequestration hold higher values, as syngas is rich in CO. Carbon monoxide dehydrogenase enzyme catalyzed the interconversion of CO to $CO_2$ reversibly [67]. Researchers have reported the CO$-$dehydrogenase complex (CODH) from *Methanosarcina thermophila* [68], *Hydrogenophaga pseudoflava* [69], *Oligotropha carboxidovorans* [70], *Thermoanaerobacter kivui* [71], etc. This enzyme is known to catalyze the assimilation reaction among CO, $CH_3$ moiety, and coenzyme-A (CoA) for the synthesis of acetyl$-$CoA [68,72]. The other one, i.e., methyl branch, represents the utilization of $CO_2$ via a multistep pathway that generates methyl moieties. $CO_2$ is assimilated into formate by formate dehydrogenase and successively reduced to a methyl group. Methyl radicle is bound to the tetrahydropterin coenzyme and then assimilated to acetyl-CoA together with CO [73].

Under aerobic conditions also, some of the microorganisms like chemolithoautotrophs and aerobic carboxydotrophs possess the potential to utilize carbon monoxide as the sole source. For chemolithoautotrophs, CO acts as a sole source of carbon, as well as electrons, and in the presence of carbon monoxide, dehydrogenase enzyme CO oxidizes to $CO_2$ [29]. While for aerobic carboxydotrophs, CO is oxidized by utilizing molecular oxygen. From here, carbon monoxide can be sequestered via the pentose phosphate pathway [28].

These pathways are themselves under the expression of related genes; hence, genetic recombination and overexpression of genes would enhance the tolerance of microorganisms and fermentation rate. Insilico analysis of *C. ljungdahlii* of the genome was conducted by genome-scale reconstruction and the OptKnock computational framework to identify gene knockouts, followed by metabolic engineering for the overproduction of ethanol, and the production of native products like butanol and butyrate along with the increase in the production of native products including lactate and 2,3$-$butanediol. In the native strain *C. ljungdahlii* iHN, the 637 genes identified contributed to 698 metabolites and 785 reactions. The OptKnock-derived strategies were combined with a spatiotemporal metabolic model considering the syngas bubble column reactor to overcome the drawbacks of decreased growth. The two-stage methodology fabricated a new *C. ljungdahlii* engineered strain which has increased product synthesis under realistic syngas fermentation conditions with the supply of CO and $H_2$ uptake rates of 35 and 50 mmol/gDW/h. The analysis also revealed that only 201 genes and 331 reactions were critical among 637 genes and 785 reactions [74]. With an aim of higher butanol and ethanol production, four genes, i.e., *adhE2, aor,* and *fnr* were considered for metabolic engineering. These genes were dedicated for acetyl$-$CoA/butyl$-$CoA to acetaldehyde/butaldehyde and acetaldehyde/butaldehyde to ethanol/butanol (*adhE2*), acetate/butyrate to acetaldehyde/butaldehyde (*aor*), and regeneration of NADH at the cost of $FdH_2$ (*fnr*), respectively. Cells were engineered with *Escherichia coli* as a donor as well as host, i.e., DH5α as host for plasmid amplification and CA434 as a donor cell for conjugation-assisted transfer. Overexpression of different genes has affected the metabolism differently, but all the engineered strains have higher alcohol production from glucose than the wild. In the presence of the *aor* gene, strains were able to reassimilate $CO_2$ even during heterotrophic growth. '*adhE2*'-overexpressed strains have ~50% higher ethanol production, while the combination of *adhE2* and *fnr* improved the butanol production by ~18% as well as ethanol by ~22%. The strains exhibiting higher alcohol production were able to re-assimilate acid [75].

Lauer et al. used a metabolic engineering approach for the overproduction of butanol and hexanol. A gene cluster of 17.9-kb carrying 13 genes from *C. acetobutylicum* (for hexanol) and *C. kluyveri* (butanol) was inserted into *C. ljungdahlii* via conjugation resulted in a butanol yield of 1075 mg/L (butanol) and 133 mg/L (hexanol) from fructose. In the case of a gaseous substrate (comprised of 80% $H_2$ and 20% $CO_2$), the yield of butanol and hexanol were 174 mg/L and 15 mg/L, respectively. The insertion of the gene cluster expressed all 13 enzymes encoded by the cluster. In the next step, a first-round selection marker was eliminated using CRISPR/Cas9, and a 7.8 kb gene cluster with 6 genes from *C. carboxidivorans* was further inserted that resulted in the hexanol and butanol titers of

251 mg/L and 158 mg/L, respectively, from the gaseous substrate. Further scale-up of fermentation to 2 L resulted in increased titer to 109 mg/L and 393 mg/L for butanol and hexanol, respectively [76]. Genetic and metabolic engineering have opened the doors to the industrial application of microorganisms with unending opportunities. It not only improved the efficiency of native fermentation products but also added new products to the metabolites.

**Table 2.** Major efforts for syngas utilization for the production of various microbial products.

| Microorganisms | Syngas Source | Products | Condition | Working Volume | Product Yield | References |
|---|---|---|---|---|---|---|
| *Clostridium butyricum* | Biomass gasification (70% forest residue + 30% charcoal), composition 13.05% $H_2$ + 22.92% CO + 7.90% $CO_2$ + 1.13% $CH_4^+$ 45.58% $N_2$ + 9.42% other gases. | Ethanol | Tar free bioreactor in fed-batch system. 500 mL impinger bottle with 100 mL working volume was used with syngas supplementation: 1 L, period 24 h, at temp 37 °C with mixing rate 200 rpm | 0.1 L | 29.94 mmol/L | [45] |
| *Clostridium ljungdahlii* DSM 13528 | Lignin-rich biomass from biorefinery and beech wood gasification at 850 °C | Acetate Ethanol | Temp 37 °C; pH 5.9, mixing rate 800 rpm; flow rate 18 mL/min (beech wood syngas) and 23 mL/min (lignin) | 1.5 L | Beech wood Acetate: 15.81 g/L, Ethanol: 0.93 g/L, Lignin Acetate: 14.88 g/L, Ethanol: 1.86 g/L, | [22] |
| *Acetobacterium woodii* | 40% $N_2$, 25% $CO_2$, and 35% $H_2$ | Acetic acid Acetone Isopropanol | Flow rate 10 mL/min, pH 7.5, temp 30 °C, mixing rate 250 rpm | 1.3 L | Acetic acid: 438 mM; acetone: 7.6 mM; and iso-propanol: 14.5 mM | [77] |
| *Clostridium aceticum* DSM 1496 and *Clostridium kluyveri* DSM 555 | - | n-caproate | Temp 30 °C, syngas feed 10 mL/min, pH 7.5, inoculum size 10%. | 1.2 L | 8.2 g/L | [78] |
| *Clostridium* sp. AWRP | 20% $CO_2$ and 80% $H_2$ at 100 kPa for 24 h 100% CO with 50 kPa | Ethanol and 2,3-butanediol | Synthetic medium, temp 37 °C, mixing rate 500 rpm, pH 5.5 aeration 0.1 vvm | 1 L | Ethanol 232 mM, 2,3-butanediol 23 mM | [79] |
| *Clostridium ragsdalei* | Pressure: 200 mbar $H_2$; 600 mbar CO, 200 mbar $CO_2$ | Ethanol | Temp 37 °C and mixing rate 100 RPM Period 44 h | 1 L | 7.67 g/L | [80] |
| Carboxydotrophic microbiome | 35% CO; 30% $CO_2$; 20% $H_2$ and 13.3% $N_2$ | Acetic acid and ethanol | Inoculum size 10% 0.1 M phosphate buffer pH 7.0 | 0.85 L | Acetic acid 3.8 g/L and ethanol 0.2 g/L | [81] |
| *Clostridium carboxidivorans* and *Clostridium autoethanogenum* | 34.4% $N_2$, 30.3% CO, 22.2% $H_2$, 9.4% $CO_2$ with traces of $CH_4$, $NH_3$ HCN, $H_2S$ | Ethanol | Period 144 h, Temp 37 °C, pH 6.0 | 1 L | Ethanol 3.60 g/L and 2,3-butanediol 1.06 g/L | [82] |
| *Clostridium carboxidivorans* P7 | 70% CO, Ar 30% (150 kPa) | Hexanol | Temp 30 °C, pH 6.0, mixing rate 100 rpm | 0.20 L | 5.06 g/L (oleyl alcohol addition), 8.45 g/L (ethanol addition) | [83] |
| *Clostridium carboxidivorans* | 80% CO and 20% $CO_2$ | alcohol | pH 6.0, Temp 37 °C flow rate 4.0 L/h CO (800 mbar) and 1.0 L/h $CO_2$ (200 mbar) | 1 L | Ethanol 1.17 g/L, Butanol 0.56 g/L, Hexanol 0.16 g/L | [84] |

Syngas fermentation into bio-alcohol: Syngas fermentation is based on the microbial utilization of gases and sequestration into value-added products and microbial biomass. Its kinetics and process dynamics are different from the conventional submerged fermentation as it solely relies upon the solubility of the gaseous phase into medium followed by its utilization [68]. Among all the microbial products, alcohol, especially ethanol, has its own market as it can be used in beverages and healthcare, besides fuels. Maintenance of anaerobic conditions during the production phase is the major prerequisite for alcohol/ethanol production. Syngas fermentation itself creates a similar environment without any preprocessing and pretreatment of feedstock (as in the case of biomass) [69]. Richter

and colleagues reported that ethanol production by *C. ljungdahlii* from syngas fermentation was mainly controlled and regulated by thermodynamics and not by the expression of responsible enzymes. Acetogenic bacteria exhibited two different physiological phases: the growth phase (acidogenesis for acid/acetate production) and the starvation phase (solventogenesis for ethanol production). Both stages were maintained at equilibrium for acetate and ethanol production, respectively, in a sequential bioreactor. Protein profiling with around 2000 proteins from both stages revealed that nutrient-limited conditions are responsible for the transition from acidogenesis to solventogenesis without any change in the enzyme population in the central energy metabolism, as enzymes responsible for ethanol production were remain present in abundance even during acidogenesis. Thermodynamic modeling recognized reduced cofactors and acetic acid as saturation reactants as well as switching points. As soon as intracellular undissociated acetic acid touched the threshold limit, bacterial cells diverted the surplus reducing equivalents for ethanol production. During syngas fermentation, reducing equivalents cannot be diverted for biomass production, and the CO-rich syngas supply was still high. Nutrient availability and pH can be used as switching points and can aid in attaining the desired level of solventogenesis [85].

Besides adaptability, the lower yield is another challenge for the commercialization of syngas fermentation as industrial gaseous discharge is comprised of inorganic carbon as $CO_x$. In order to improve the gaseous fermentation and ethanol yield, a mixed gas stream containing CO, $CO_2$, and $H_2$ was continuously aerated to culture along with supplementation with rabbit faeces. The addition of rabbit faeces enriches the ethanol-producing strains by a change in the microbial community composition with *Blautia* as the dominant strain (41.1% of community structure). Post-adaptation, around 14.07% $CO/CO_2$ sequestered to organic carbon, and ethanol productivity reached 1.41 g/L [86]. The main reason for using faecal matter is the enrichment of responsible microorganisms that contribute to syngas fermentation and product yield. The faeces from herbivores like rabbits and horses [87] are usually rich in homoacetogens, which are considered efficient microorganisms for syngas fermentation. The addition of faecal matter increased the population of homoacetogens and thus improved syngas fermentation. Fernández-Blanco and colleagues combined syngas fermentation and chain elongation for the synthesis of bioalcohol and medium chain length fatty acids (C6–12) like n-caproic acid and n-butyric acid. However, pH is the major hindrance in this approach; thus, *C. aceticum* and *C. kluyveri* were co-cultured around neutral pH with a continuous syngas supply. Co-culture systems offered n-butyrate and n-caproate a productivity of 1.96 g/L and 1.14 g/L, respectively [78].

Kwon et al. reported the inhibitory effect of high concentrations of CO on $CO_2$ reduction to formate. Acetate supplementation mitigates CO-assisted inhibition in *Clostridium* sp. AWRP-based fermentation. Bacterial cells oxidized CO and side by side reduced the acetate into ethanol and aided in countering the inhibitory effect. The presence of acetate enhanced the specific growth rate by 83% over control. Even a bioreactor trial with exogenous acetate supply, ethanol, and 2,3-butanediol titer increased by 2.9- and 2.3-fold and attained concentrations of 232 mM and 23 mM [79]. Hexanol is a high-demand alcohol used as solvent and feed in chemical synthesis and plastic industries. Currently, the petroleum industry and chemical synthesis are the major or sole contributors. Colleagues have identified the possibility of hexanol production via syngas fermentation. Various researchers have suggested that acetogenic bacteria produce two types of $C_2$ compounds including acetate and ethanol, but the same organisms can also be used for hexanol production by varying the production conditions as in the current case hexanol concentration of 0.09 g/L which was achieved by cultivation temperature from 30 to 37 °C. In addition, it was also found that an increase in CO content in syngas supply from 30 to 70% increased the total alcohol titer to 4.32 g/L and was comprised of 1.90 g/L hexanol, 1.20 g/L ethanol, and 1.20 g/L butanol. The work emphasized that hexanol production can be increased by supplementing the medium with ethanol (2.34 g/L with 2 g/L ethanol supplement). The final hexanol productivity of 0.18 g/L.day was achieved with ethanol supplementation [19].

## 3. Opportunities and Challenges in Syngas Fermentation

Gaseous fermentation and syngas sequestration have gained much attention in research as well as at the commercial scale for the production of high-value commodity fuels and chemicals. However, the commercialization of technology needs to answer some challenges in order to achieve global acceptance and commercialization. Some of the challenges are summarized below:

### 3.1. Gasification and Syngas Cleaning

Syngas production via gasification of biomass needs critically high temperatures and is conducted in the presence of a catalyst. Usually, processes operate at higher temperatures and pressures and require specialized instruments and technical expertise. Consistency of process and composition of produced syngas are the main challenges along with catalyst regeneration, maintenance of high temperature, and processes that add costs to the whole process and affect its economics when taking the process from lab to pilot/commercial scale [88].

Biomass composition, catalysts, gasifying agents, and operational conditions are the factors that govern the syngas composition. In the context of biomass, high-lignin content in biomass resulted in syngas with higher $H_2$ content. In addition, gasification in the presence of air or $O_2$ increased the $CO/H_2$ ratio [89]. Besides CO and $H_2$, NOx, COx, tar compounds, hydrogen cyanide, methane, ammonia, hydrogen sulfide, carbonyl sulfide, tars, etc., are also present in syngas. Among these, CO, $CO_2$, and $H_2$ can be utilized while tar compounds, and NOx and SOx can hinder enzyme activity, microbial growth, and product yield. Removal of microbial inhibitors needs syngas cleaning which is a multistep and cost-intensive process. Besides syngas cleaning, optimization of feedstock type and amount and operational conditions can also minimize the formation of impurities during gasification. Hyper-producer and robust microbial strains can be used to overcome the need for synthesis gas cleaning. In addition, Sox, NOx, and other inhibitors can also be utilized [88,89].

Syngas components, i.e., CO and $H_2$, act as electron donors in fermentation and microbial metabolism. Carbon monoxide is a suitable source of electrons at a wider pH range, partial pressure of gas, ionic strength, and electron pair carrier; however, generation of reducing power from CO is not preferred as it diverts the metabolic route towards $CO_2$ and is far from ethanol or metabolic products. A higher concentration of CO reduced the conversion efficiency. However, at higher concentrations, $H_2$ induced the use of electron donors and diverted the metabolic pathway toward microbial metabolites like cell biomass and ethanol, etc. In lower $H_2$ concentrations, CO becomes the preferred electron donor but it lowered the available carbon for other product syntheses, and in higher $H_2$ concentrations, acetate production is favored over ethanol [89]. In short, even if microorganisms do not require rigorous $H_2/CO$ ratios, this is an important parameter in order to obtain the desired products and high carbon conversion efficiencies.

### 3.2. Bioreactor Design and Mass Transfer

The fermentation process relies on microbial growth and substrate availability and utilization. These parameters are prone to change with the scale of operation. Thus, mass transfer, substrate limitations, and product inhibition are the major challenges for fermentation. In the case of gas fermentation, risks are involved with the biocatalyst as an efficient microorganism mandatory for the rapid sequestration of gaseous carbon, especially on a large scale. Under anaerobic conversion, microbial culture has a lower density in comparison to aerobic due to lower ATP generation; hence, it becomes necessary to attain threshold cell concentration before entering the production phase for efficient conversion and high production rates. To overcome this challenge, evaluation of each step and optimization of media composition is a must [90]. Microbial contaminants and lower productivity are among the major bottlenecks as CO and $H_2$ both are relatively insoluble, and, in syngas fermentation, CO is the sole carbon source. Stoichiometric analysis for

ethanol production from carbon monoxide revealed a 6:1 substrate-to-product ratio [91]. In such conditions, efficient mass transfer becomes very important along with enhancing the solubility of gases [88].

A similar observation was reported by Wan et al. while working on *Clostridium carboxidivorans* P7 for industrial syngas fermentation. Multiple carbon fixation routes including anaplerotic pathways, pyruvate: ferredoxin oxidoreductase reactions, and the Wood-Ljungdahl pathway have been identified for carbon sequestration. The Wood-Ljungdahl pathway has a higher flux, but the anabolic pathways' activity was very low. Even in a bioreactor culture system, extracellular products (acids and alcohols) remain the same even on an increasing syngas flow rate from 1 to 10 mL/min. However, alcohol accumulation increased but further increased in flow rate to 20 mL/min, alcohol productivity did not improve mainly due to mass transfer limitation [92]. Reactor design can play a crucial role and might improve fermentation by enhancing mass transfer as well as overcoming the solubility issue. Previous research has suggested that, in comparison to conventional models, additional efforts must be needed to fabricate an efficient cultivation system. Instead of mechanical mixing, gaseous fermentation with a membrane system or modified gas supply offers a higher yield that increases the chances to diffuse through the cell membrane and can be utilized. For fermentation, the selection of the membrane is itself a challenge as they are symmetric, porous, and asymmetric membranes. All of them have advantages and disadvantages. For example, a system with higher permeability might prefer an asymmetric membrane, but the shape and configuration of the membrane must also be checked for mass transfer and efficient conversion [93]. Roy and colleagues modified the gas inlet system in a continuous stirred tank bioreactor system. The aeration tube was fitted at the bottom throughout the periphery of the reactor and has multiple discharge points. The effluent was taken off from the top of the reactor. The system works with mechanical stirring. Multi-point inject discharge adds more gas to the reactor and mechanical mixing provides lateral movement to bubbles which increased the pathway traveled by them. It increased the interaction of gas with liquid and cells and thus utilization and yield increased with the mixing rate [94].

Sathish and colleagues developed a unique bulk-gas-to-atomized-liquid (BGAL) contactor to improve mass transfer via liquid atomization to discrete droplets. Atomization increased the liquid and bulk gas interface. The bulk-gas-to-atomized-liquid (BGAL) contactor attained a transfer rate of 569 mg/L·min (KLa: 2.28 /s) when operated with oxygen as a model. For syngas fermentation, the BGAL contactor was combined with a packed bed. The combination system prevents the dispersion of gas-saturated droplets into the bulk liquid and dilutes the available gas in the liquid phase. It offered a KLa of 0.45–1.0/s which is lower than the contractor alone but still 20 times higher than the stirred tank reactor. In terms of energy, the contactor-packed bed bioreactor reduced the energy requirement by four-fold compared to the stirred tank reactor. Enhanced mass transfer improved the syngas to ethanol conversion by *Clostridium carboxidivorans* P7 and ethanol productivity reached 746 mg/L·h (ethanol/acetic acid: 7.6) which was around two-fold higher than available reports [95].

Microbial contaminant competes with the main microbial strain for carbon source, energy, and nutrients. As seen during acetone-butanol-ethanol (ABE) fermentation, bacteriophage contamination changes bacteria cell morphology, slows fermentation rates, and lowers product yields. While adapting the concept of gas fermentation, molasses is replaced with gas and operated with a minimum nutrient supply, and the possibility of bacteriophage and other microbial contaminants is eliminated in the gaseous phase due to the high temperature [88].

### 3.3. Downstream Processing

Downstream processing is a sequential multistep process to eliminate contaminants and recover the target product with high purity. Based on the process and target product, each method has already been optimized and validated which can be adopted as required.

Usually, there is no technical obstacle faced in fermentation downstream. In the case of syngas/gaseous fermentation, the microbial metabolite/product is in low concentration; thus, downstream becomes more energy and cost-intensive. The cost and energy efficiency can be improved if downstream is operated in a closed loop system along with recycling process residues [88].

Large-scale operations, especially running on a continuous mode, require regular removal of products from the system along with maintenance of high cell densities within the reactor. Generally, the concentrations of solvents including ethanol, butanol, acetone, etc., in gas fermentation lie within 6–8% [*w/v*] which makes regular recovery costly and energy intensive. In addition, microbes are also sensitive to these solvents, so productivity is also affected. Distillation and other vacuum operations are effective but energy-intensive [96,97]. Some alternate strategies or improved systems are required for product recovery with lower costs. Liquid–liquid extraction is one of the methods in which aqueous and nonaqueous phases (solvents) are used to separate the solute based on their respective solubilities.

Tsenang and colleagues studied liquid–liquid extraction (LLE) for simultaneous detection and separation of ethanol from home-brewed alcoholic drinks, collected from local brewers with high accuracy and repeatability. The liquid–liquid extraction (LLE) studies were conducted with solvents including ethyl acetonitrile, acetate, dichloromethane, and hexane. Among them, ethyl acetate (3:1) efficiently recovers the ethanol (93.48%) from the sample which was much higher than other solvents [98]. Lee and colleagues developed a process for the recovery of ethanol using 2-methyl pentanol as a solvent. Thermodynamic modeling, liquid–liquid equilibrium (LLE) with molecular simulations, and experimental validation reduced the heat duty by one-third under optimum conditions over the control recovery processes [97]. In liquid–liquid separation, water-insoluble organic solute/products get separated from fermentation broth to the nonaqueous/organic phase. However, some loss of byproducts and feedstock might also occur during the recovery process. Additionally, the frequent use of solvents also affects microbial growth due to toxicity, formation of emulsion, etc. The creation of a physical barrier is another approach that aids in product recovery known as perstraction [96].

Huang et al. used a supported ionic liquid membrane (SILM) for ethanol recovery from dilute aqueous solution by perstraction with a comparative evaluation of three trihexyl (tetradecyl) phosphonium ionic liquids as extracting solvents including [THTDP][Br], [THTDP][N(CN)$_2$], and [THTDP][Tf$_2$N], and three polymeric microporous membranes: hydrophilic Durapore® GVWP, hydrophobic Durapore® GVHP, and polypropylene (PP). Among the solvents and membranes compared, [THTDP][N(CN)$_2$]-GVHP SILM offered maximum extraction along with stability. The functionality of the system relies on the type and load concentration. With 2 wt% feed (ethanol), SILM was functional for ~240 h without intermixing, and it can withstand a high ethanol flux (>2.2 kg/m$^2$·h) which was much higher than pervaporation (~$10^{-4}$–$10^{-1}$ kg/m$^2$·h) and selectivity was also high at > 320, in comparison to liquid–liquid extraction (~20). The recovered ethanol attained 80% purity and selectivity of 200 by single-stage vacuum distillation [99]. This method offers an advantage as there is no toxicity imposed on microbes due to the extractant also suffering from a slow rate of separation as well as membrane fouling [96].

Pervaporation is a product recovery technique in which fermentation broth passed through a membrane that selectively removes volatile compounds. During operation, a vacuum or partial pressure gradient is maintained across the membrane that facilitates the vaporization of volatile compounds including ethanol, which is then collected on another side by condensation [96].

Production of sodium pyruvate is also accompanied by ethanol and water. The byproducts are discarded in the residual mother liquor. Two-step pervaporation-crystallization was employed for the simultaneous recovery of sodium pyruvate and ethanol from the residual mother liquor. The first step operates at a low temperature that removes water fraction from the liquor followed by cooling crystallization in the second step. The sample was collected and processed via NaA zeolite membranes for ethanol-sodium pyruvate sep-

aration which offered a high water/ethanol separation factor (>10,000). However, regular use and high load led to pyruvate deposition on the outer membrane surface which lowered the total flux. In addition, the membrane can be regenerated merely by water washing [100]. Like conventional fermentation and other commercial processes, syngas fermentation also generated some residues and faces the challenge of a substrate utilization ratio. As in gaseous fermentation, the feedstock/main substrate is gas, so residual gas sparging can be done to improve substrate utilization. Exit gas stream from bioreactor or after product recovery recycled and injected back into the bioreactor. Gas stripping in sugar-to-ABE fermentation by *C. beijerinckii* mutant BA101 not only led to complete substrate utilization with complete acid-to-solvent conversion but also improved productivity by 200% [96].

## 4. Techno-Economic Analysis

Gaseous discharge is a common problem for almost all industrial processes like electricity, power, transport, textiles, chemical industries, etc., that can be used after processing/cleaning by microorganisms and sequester oxides of carbon, and nitrogen mainly to biomass and other products. However, the cost of the process is one of the major concerns due to gas cleaning and the extent of utilization. Higher conversion and tolerant microorganisms might lower the cost of the conversion process as well as the primary process.

LanzaTech (NZ/US-based company) utilized gaseous discharge from industries especially steelmaking as feedstocks for bioethanol and 2,3-butanediol production using Clostridial biocatalyst with minimal gas conditioning. Reutilization and sequestration of gaseous discharge make the steelmaking process greener by reducing the carbon footprint. Based on this technology, LanzaTech has scaled up the production of bioethanol to 100,000 gallons per year in Shanghai, China with its partner Baosteel Group [96]. Comparative evaluation of conventional fermentation and syngas fermentation found syngas fermentation superior over conventional modes due to its lower requirements of pretreatment and energy. However, the technological aspect is still a challenge for this.

Biomass gasification generates oxides of carbon and $H_2$ that were used for poly-hydroxyalkanoates (PHA) production by Choi et al. [101]. Switchgrass was processed thermo-chemically to generate syngas, mainly comprised of $H_2$ and CO, and fermented using *Rhodospirillum rubrum* to sequester it to PHA. Fermentation yielded 12 mg PHA and 50 mg $H_2$. The overall evaluation showed a grassroots capital investment of $ 55 million and $6.7 million in annual operating costs. The total product cost for $H_2$ and PHA was calculated to be $2.00/kg for $H_2$ and $1.65/kg PHA, respectively, [101] which was much lower than the actual minimum selling price (MSP) of PHAs, i.e., 2.41–4.83 $/kg [102]. Bioethanol is a common conversion product from syngas fermentation which has a lower cost, as no pretreatment of biomass or hydrolysis is required. Integrated/combined processes have better results in comparison to isolated processes. Techno-economic analysis of integrated systems comprised of biomass gasification-syngas fermentation-solid oxide fuel cell systems for ethanol production suggested that a dual fluidized bed with woodchips offered higher CO and $H_2$ content. Additionally, the substrate conversion rate increased ethanol yield. A tentative production cost was estimated to be 430.1 €/t ethanol which can be lowered down to 64.5 M€ (total product price) with attainment of system efficiency of 55% by improving syngas utilization in the system, its distribution, recycling of fermenter exhaust by solid oxide fuel cell systems and by using the 'Technique for Order of Preference by Similarity to Ideal Solution' method [103]. Organic waste material can be used as a mineral or nutrient supplement that is utilized by microorganisms for growth and product formation. Lee and colleagues have utilized sewage treatment plant sludge filtrate for acetate production by *Eubacterium limosum* KIST612. It was reported that the use of sludge filtrate instead of yeast extract not only reduced the medium cost by 84% but also increased the acetate titer by 130% [104].

The performance of a hybrid plant, working for synthetic natural gas production from biomass, as well as carbon capture and storage, was assessed using simulation-based analysis. The plant was utilizing 6.25 dt/h of virgin biomass yields and produced 1.32 t/h of synthetic natural gas. The energy efficiency of the plant was 51.2% and stored 2.97 t/h of $CO_2$. Simulation-based analysis suggested that the synthetic gas process becomes economical if it achieves the target of 92.14 £/MWh and with an integrated approach for larger operations, the price was reduced by about 15% [105]. A hybrid approach was adopted for ethanol production from bagasse via biomass gasification followed by syngas fermentation. Aspen plus and MATLAB simulated the utilization of biomass for an ethanol yield of 283 L/per dry tonne of bagasse along with 43% energy efficiency. The simulation was targeted for multiobjective optimisation to improve energy efficiency and reduced costs. Considering typical discounted cash flow, the analysis lead to a minimum ethanol selling price of 0.69 $/L [106]. Medeiros et al. evaluated the performance of a bubble column reactor for syngas fermentation followed by anhydrous ethanol recovery using artificial neural networks and a multi-objective genetic algorithm. The major factors considered for simulation and optimization were the investment required, the energy efficiency of a process, the minimum selling price, and, most importantly, the bioreactor productivity that influences the whole economics of the process. With integrated production and recovery processes, the approach achieved a production rate of 124–133 MML/year. The calculated MSP was 0.707–0.713 $/L with high mass transfer [107]. Different strategies have been evaluated for syngas fermentation and ethanol production along with methane from crude glycerol. The study group has been categorized into three: bioethanol production without $CO_2$ capture, bioethanol production with $CO_2$ capture, and bioethanol with $CO_2$ capture followed by biomethanation of captured $CO_2$. For fermentation, syngas was produced by biomass gasification. The working volume for glycerol utilization was 50,000 metric tons per year. The techno-economic feasibility suggested that groups have energy efficiency within the range of 30.2% to 35.1% among which group 3 has the highest i.e., 35.1%. MSPs for bioethanol were reduced to $1.32/L (group 1), $1.4/L (group 2), and $0.31/L (group 3) [108].

The studies also emphasized the same fact that integrated systems have higher profitability and minimum production costs as also suggested by Bhatia et al. [109]. Most of the research has shown that industrial process discharge residues and waste in different forms like gases and solid and liquid effluents [110–113]. These wastes are usually rich in carbon, nitrogen, phosphate, sulfur, organic compounds (proteins, carbohydrates, lipids, fatty acids, lignin, phenols, cresols, etc.), and inorganic salts oxides ($CaCO_3$, $CO_x$, $SO_x$, and $NO_x$). The accumulation of these wastes increases the environmental load and deteriorates it. The possible reutilization of process waste including gaseous discharge for the production of some valuable products via a secondary process improves not only the process economics of the primary process but also its environmental feasibility. The process relies on biocatalysts whose performance is governed by physical and chemical growth conditions. The secondary process also needs optimization and upgradation like other chemical/biological processes. The syngas fermentation approach has shown a promising future for the existing industrial and commercial processes to make them greener and more efficient along with an increase in the production of a range of valuable products [14,109].

## 5. Conclusions

Syngas offers an opportunity for microbial conversion to value-added products via a greener process. Chemical processes are somehow efficient but operate at typically high temperatures and pressures. In addition, the chemical process also utilizes toxic solvents and costly catalysts. Microbial organisms also act as a catalyst for the sequestration of CO as a carbon source and synthesize various metabolites via microbial metabolism like ethanol, microbial biomass, biodiesel, higher alcohol, etc. In comparison to conventional fermentation with lignocellulosic biomass, syngas/gas fermentation offers an advantage as no biomass hydrolysis is needed for feed/hydrolysate preparation. Hence, the feedstock

becomes much cheaper for cost-effective product synthesis. In the view of the industrial aspect, the process operates at mild operational conditions and has a lower cost in comparison to conventional fermentation. In terms of the environment, the process offers a way to transform the waste gases from industrial processes and fuel burning along with revenue generation. It might contribute to making an industrial process eco-friendly and economical for large-scale production.

**Author Contributions:** V.A.; Conceptualization, methodology, writing—review and editing. A.K.B.; investigation, resources. B.R.; writing—review and editing, Y.-H.Y.; supervision, writing—original draft preparation. S.K.B.; Conceptualization, methodology, writing—review and editing, supervision. All authors have read and agreed to the published version of the manuscript.

**Funding:** This research was supported by the C1 Gas Refinery Program through the National Research Foundation of Korea (NRF), funded by the Ministry of Science and ICT (2015M3D3A1A01064882), and by the National Research Foundation of Korea (NRF) [NRF-NRF-2022M3I3A1082545, NRF-2022M3J4A1053702, NRF-2022R1A2C2003138 and NRF-2021R1F1A1050325].

**Institutional Review Board Statement:** Not applicable.

**Informed Consent Statement:** Not applicable.

**Data Availability Statement:** Not applicable.

**Acknowledgments:** The authors acknowledge the KU Research Professor Program of Konkuk University, Seoul, South Korea and Chandigarh University, Punjab India.

**Conflicts of Interest:** The authors declare no conflict of interest.

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
