# Peer review of "A Mini-Review on Syngas Fermentation to Bio-Alcohols: Current Status and Challenges"

_sustainability, doi:10.3390/su15043765_

Round 1

Reviewer 1 Report

General comments:

In this manuscript (Submission ID: sustainability-2184819), the author reviews the recent trends in syngas fermentation for the production of ethanol and other energy-related products. In order to overcome the defects of traditional chemical methods, such as high energy consumption, harsh reaction conditions and toxic by-products, the application of microorganisms in syngas production is put forward. However, the manuscript was not well presented, and there were too many drawbacks and unclear points, which should be improved. There are some problems that must be solved before it is considered for publication.

1.    In line 45, the author makes this point: "Reduced soil fertility and nutrient cycling are the second major ill effect of air pollution." Please tell me which reference this is from or the author's original.

2.    Please could you clarify where Fig. 1 and Fig. 2 in the paper have been obtained from? If these figures were not drawn by you or one of your co-authors, please indicate the references in the figure caption.

3.    I did not see the author's comments with a unique personal perspective about the topic of this article. Just the reorganization of results does not result in progress in the scientific community. For example, in lines 518-519, the author mentions that "However, it also needs optimization and upgradation like other chemical/biological process.". I hope the author can further put forward his own views on how to optimization and upgradation the existing process.

4.    In line 142, note the format of the flow rate unit. Please check the full text carefully to avoid other similar problems, like 233.

5.    In line 333, no space between the figure (30-37) and the ℃.

6.    Please note the format of the References. I have noticed that the format of the references is not uniform, such as the references [56] and [64], and similar problems exist in other References. Please check them carefully. 

Author Response

Response to reviewer # 1

Reviewer #1

R1Q1:  In line 45, the author makes this point: "Reduced soil fertility and nutrient cycling are the second major ill effect of air pollution." Please tell me which reference this is from or the author's original.

R1A1: We appreciate reviewers’ effort and comments to improve the standard of our manuscript. This is the observation from previous researches and trends. However, the fact is also have scientific proof and hence the relevant literature have been cited in the text (line 45-50; reference 2, 5-9).

R1Q2:    Please could you clarify where Fig. 1 and Fig. 2 in the paper have been obtained from? If these figures were not drawn by you or one of your co-authors, please indicate the references in the figure caption.

R1A2: The figures present in this manuscript were self-designed.

R1Q3:   I did not see the author's comments with a unique personal perspective about the topic of this article. Just the reorganization of results does not result in progress in the scientific community. For example, in lines 518-519, the author mentions that "However, it also needs optimization and upgradation like other chemical/biological process.". I hope the author can further put forward his own views on how to optimization and upgradation the existing process.

R1A3: We have added our opinion and observation at several points in the text and improved the manuscript. Especially the point raised for optimization and upgradation…. Has been elaborated. (Page 21; Line 620-666).

‘Most of the research have shown that industrial process discharge residues and waste in different forms like gases, solid and liquid effluents. These wastes are usually rich in carbon, nitrogen, phosphate, sulfur, organic compounds (proteins, carbohydrates, lipids, fatty acids, lignin, phenols, cresols, etc.), and inorganic salts oxides (CaCO3, COx, SOx, NOx). Accumulation of these wastes increases the environmental load and deteriorates it. The possible reutilization of process waste including gaseous discharge for the production of some valuable products via secondary process improves not only the process economics of the primary process but also its environmental feasibility. The process relies on biocatalysts whose performance is governed by physical and chemical growth conditions, the secondary process also needs optimization and upgradation like other chemical/biological processes. Syngas fermentation approach has shown a promising future for the existing industrial and commercial processes to make them greener and more efficient along with an increase in the product of a range of valuable products.’

R1Q4:    In line 142, note the format of the flow rate unit. Please check the full text carefully to avoid other similar problems, like 233.

R1A4: We have gone through the manuscript and all the units have been corrected properly.

R1Q5:    In line 333, no space between the figure (30-37) and the ℃.

R1A5: The uniform pattern has been updated for all the units and digits.

R1Q6:    Please note the format of the References. I have noticed that the format of the references is not uniform, such as the references [56] and [64], and similar problems exist in other References. Please check them carefully. 

R1A6: We have checked all the references and updated them with Zotero reference manager and all the references were corrected as per journal guidelines.

Reviewer 2 Report

"A mini-review on syngas fermentation to bio-alcohols: current status and challenges" is of interest of the journal's readers, since the topic is a promising opportunity to utilize syngas from various sources to produc value-added products. Nevertheless, the manuscript has to be improved by major revisions before it should be accepted:

- Please add a literature review in the introduction, which illustrates, which reviews can be found in literature already e.g. on various biofule production pathways covering syngas fermentation as well, and outline why this manuscript is necessary to complement existing literature (see also point regarding structure and USP below).

- The introduction should provide a more holistig view on the problems of biomass burning and fossil fuel usage. 

- The main part itself, summarizing existing literature, has to be improved in regard of value-added and USP of this mini-review. The scientific depth of the description of all used literature sources is quite low. It would be better to focus on one aspect of syngas fermentation and elaborate in more detail on that. This copuld be e.g. necessary syngas qualities, fermentative pathways, metabolic engineering, process challenges like reactor designs, techno-economic analyses. Furthermore, to provide additional value to the already existing articles, it is necessary to contextualize the findings of these scientific papers and reframe, compare or discuss them critically. This ensures, that the proposed manuscript provides additional insights and represents its USP.

- Some sections, e.g. syngas production, only discuss specific setups and only a few possible, technical options. Please ensure, when you want to cover all aspects of syngas fermentation, to provide a comprehensive picture of all technical approaches (e.g. gasificartion technologies, other syngas sources as Co-SOEC,...). Furthermore, it is not comprehensive, why certain literature sources were discussed in the review, e.g. in Table 2, and how this selection represents an extensive overview of the existing research results.

- The language and form has to be improved at certain points to meet scientific standards and make sure, that the messages of the sentences are understandable. (no slashes and &, Eq 1, "wiz" in line 75, line 104-109, "too" line 521...). Figure 1 does not illustrate all possible pathways and gives only an overview of one specific setup, whereas its additional information value is not clear. 

- Please make sure, that all used literature is cited properly and all information from other sources are marked as citations, e.g. Fig 2, line 484-492,...

- A more alborate discussion of the techno-economic results in literature is necessary. When comparing different obtainable prices, it is important to state the relevant assumptions for the calculations (e.g. assumed plant size) to enable proper comarisons. I suggest adding an extensive table, comparing a broader range of literature results and giving more details about the assumptions.

- The conclusion section has to be improved significantly. What is the conclusion of the extensive literature review? (see "USP")

Author Response

Response to reviewer # 2

Reviewer #2

R2Q1: Please add a literature review in the introduction, which illustrates, which reviews can be found in literature already e.g. on various biofuel production pathways covering syngas fermentation as well, and outline why this manuscript is necessary to complement existing literature (see also point regarding structure and USP below).

R2A1: We appreciate the reviewer’s suggestions to improve the standard of our manuscript. We have discussed recent literature related to syngas fermentation into biofuels and also added points related to the importance of this manuscript. Page 3; line 91-100. Microorganisms have shown potential for the utilization of CO as a carbon source and/or electron donor followed by its utilization via various metabolic pathways i.e. anaerobes follow the ‘Wood-Ljungdahl pathway’ [27] and sequester it into acetyl-CoA while under an aerobic environment carboxydotrophs used molecular oxygen for its oxidation to CO2 followed by sequestration via pentose phosphate [28]. There is another group, chemolithoautotrophs which oxidized CO by carbon monoxide dehydrogenase to CO2 [29]. For all these pathways, acetyl-CoA is the junction point and its further utilization depends upon the growth conditions prevailing. The pathways determining the end products are under the control of gene expression hence genetic modification might contribute in improving the fermentation rate and product formation.’

Page 3 Line no 101-111

Syngas fermentation represents an opportunity to transform waste gases into commercially viable products rather than leaving them to deteriorate the environment. The available literature also stated that inorganic as well as organic carbon can be converted into commodity products [30,31]. Very few articles have focused on the production of energy-related products from syngas via microbial fermentation. In the current review, recent trends in syngas fermentation for the production of ethanol and other energy-related products are summerized (Figure 1). Various parameters that affect the syngas fermentation process are discussed and techno-economic aspects are also evaluated for industrial as well as social acceptance of the process. The integration of processes with the currently available industrial process is emphasized to minimize pollution and make processes greener and more cost-effective.

R2Q2: The introduction should provide a more holistic view on the problems of biomass burning and fossil fuel usage.

R2A2: We have dedicated the first three paragraphs in the introduction for environmental pollution and ill-effect and issues related to biomass burning and fossil fuel usage are also discussed in this. (Page 1; Line 34-63).

Increased energy requirements and fuel consumption have raised environmental pollution and lowered air quality. During the inter-season period, burning agricultural residues (left out after harvesting) is a common practice that not only releases pollutant gases but also aerosols [1]. It has been found that biomass burning contributed around 10-70% of PM 2.5 and with respect to health PM 2.5 and PM10 are among the most lethal pollutants [2]. Although, biomass burning contributes significantly but it is not the main source of air pollution. Transportation, industries operations including construction, power plants, and indoor emissions are also among the top air polluters [3,4]. Industrial operations alone contribute 23% SO2, 15% CO and 14% PM 2.5. On the other hand, transportation contributed 53% NOx, 18% CO, 17% PM2.5, and 13% PM 10 while stubble burning shared only 14% CO and 12% PM 2.5 [2].

Reduced soil fertility and nutrient cycling are the second major ill effect of air pollution [5–7]. Stubble burning raised the temperature of the soil and eliminates beneficial microorganisms. Reduction in microbial diversity affects the nutrient cycle [8,9] and the availability to plants which ultimately deteriorates agricultural productivity. Besides agricultural productivity, air pollution also leads to injury to leaves and grains, affects metabolism and enzyme activity, and promotes discoloration, chlorosis, and necrosis [2]. Air pollution exposure leads to skin and eye irritation, respiratory distress, neurological, and cardiovascular disorder. Chronic exposure can lead to permanent health ailments like respiratory disorders, Chronic Obstructive Pulmonary Disease, and even cancer [2,10].

Air pollution not only leads to biodiversity loss, and global temperature rise but also to the loss of GDP. An increase in global temperature by 3.2 °C can consume 18% of the worldwide GDP [11]. To control the financial loss and deterioration of the environment, Paris agreement targets climate change and limiting the global temperature rise by controlling GHG emissions well and a standard temperature rise has been set to keep it within 2 °C (1.5 °C preferred) [12]. Intergovernmental Panel on Climate Change suggested a preferred global warming level to 1.5 °C which can be accomplished only by lowering the global CO emission by at least 45% by 2030 and subsequently to net-zero emission by 2050. Transportation and industrial sectors have been identified as prime GHGs emitters comprising 16 and 27% of the total emissions [11].

R2Q3: The main part itself, summarizing existing literature, has to be improved in regard of value-added and USP of this mini-review. The scientific depth of the description of all used literature sources is quite low. It would be better to focus on one aspect of syngas fermentation and elaborate in more detail on that. This could be e.g. necessary syngas qualities, fermentative pathways, metabolic engineering, process challenges like reactor designs, techno-economic analyses. Furthermore, to provide additional value to the already existing articles, it is necessary to contextualize the findings of these scientific papers and reframe, compare or discuss them critically. This ensures, that the proposed manuscript provides additional insights and represents its USP.

R2A3:  Thank you for your valuable comments. Scientific depth has been improved by adding new information at several points. This review article is only focused on syngas fermentation into bioalcohols. Further, the content has been updated with the addition of some more data to make every section scientifically sound.

Metabolic engineering: Page 10-11; line 304-343

These pathways are themselves under the expression of related genes hence genetic recombination and overexpression of genes would enhance the tolerance of microorganisms and fermentation rate. Insilico analysis of Clostridium ljungdahlii of the genome was conducted by genome-scale reconstruction and the OptKnock computational framework to identify gene knockouts, followed by metabolic engineering for the overproduction of ethanol, production of native products like butanol and butyrate along with the increase in production of native products including lactate and 2,3-butanediol. In the native strain C. ljungdahlii iHN, 637 genes identified, contributed to 698 metabolites and 785 reactions. The OptKnock-derived strategies were combined with a spatiotemporal metabolic model considering the syngas bubble column reactor to overcome the drawbacks of decreased growth. The two-stage methodology fabricated a new C. ljungdahlii engineered strain which has increased product synthesis under realistic syngas fermentation conditions with the supply of CO and H2 uptake rates of 35 and 50 mmol/gDW/h. The analysis also revealed that only 201 genes and 331 reactions were critical among 637 genes and 785 reactions [70]. With an aim of higher butanol and ethanol production, four genes i.e. adhE2, aor, and fnr were considered for metabolic engineering. These genes were dedicated for acetyl-CoA/butyl-CoA to acetaldehyde/butaldehyde and acetaldehyde/butaldehyde to ethanol/butanol (adhE2), acetate/ butyrate to acetaldehyde/ butaldehyde (aor), and regeneration of NADH at the cost of FdH2 (fnr) and respectively. Cells were engineered with E. coli as a donor. Overexpression of different genes has affected the metabolism differently but all the engineered strains have higher alcohol production from glucose than the wild. In presence of aor gene, strains were able to re-assimilation of CO2 even during heterotrophic growth. ‘adhE2’ overexpressed strains have ∼50% higher ethanol production while the combination of adhE2 and fnr improved the butanol production by ∼18% as well as ethanol by ∼22%. The strains exhibiting higher alcohol production were able to re-assimilate acid [71].

Lauer et al., used a metabolic engineering approach for the overproduction of butanol and hexanol. A gene cluster of 17.9-kb carrying 13 genes from C. acetobutylicum (for hexanol) and C. kluyveri (butanol) were inserted into C. ljungdahlii via conjugation resulted in a butanol yield of 1075 mg/L (butanol) and 133 mg/L (hexanol) from fructose. In the case of gaseous substrate (comprised of 80% H2 and 20% CO2), the yield of butanol and hexanol were 174 mg/L and 15 mg/L respectively. Insertion of gene cluster expressed all 13 enzymes encoded by the cluster. In the next step, first-round selection marker was eliminated using CRISPR/Cas9 and 7.8 kb gene cluster with 6 genes from C. carboxidivorans was further inserted that resulted in the hexanol and butanol titers of 251 mg/L and 158 mg/L respectively from the gaseous substrate. Further scale-up of fermentation to 2 L resulted in increased titer to 109 mg/L and 393 mg/L for butanol and hexanol respectively [72]. Genetic and metabolic engineering have opened the doors to the industrial application of microorganisms with unending opportunities. It not only improved the efficiency of native fermentation products but also added new products to the metabolites.

Syngas quality: Page 15; Line 426-429; 439-450

Biomass composition, catalysts, gasifying agents, and operational conditions are the factors that govern the syngas composition. In context to biomass, high-lignin content in biomass resulted in syngas with higher H2 content. In addition, gasification in presence of air or O2 increased the CO/H2 ratio [85]. 

Syngas components i.e. CO and H2 act as electron donors in fermentation and microbial metabolism. Carbon monoxide is a suitable source of electrons at a wider pH range, partial pressure of gas, ionic strength, and electron pair carrier however generation of reducing power from CO is not preferred as it diverted the metabolic route towards CO2 and far from ethanol or metabolic products. A higher concentration of CO reduced the conversion efficiency. However, at higher concentrations of H2 induced the use of electron donors and diverted the metabolic pathway toward microbial metabolites like cell material, and ethanol etc. In lower H2 concentrations, CO becomes the preferred electron donor but it lowered available carbon for other product synthesis and in higher H2 concentrations, acetate production is favored over ethanol [85]. In short, even if microorganisms do not require rigorous H2/CO ratios, this is an important parameter in order to obtain the desired products and high carbon conversion efficiencies.

Bioreactor design: Page 16; Line 475-492

Reactor design can play a crucial role and might improve fermentation by enhancing mass transfer as well as overcoming the solubility issue. Previous researches have suggested that in comparison to conventional models, additional efforts must be needed to fabricate an efficient cultivation system. Instead of mechanical mixing, gaseous fermentation with a membrane system or modified gas supply offers a higher yield that increased the chances to diffuse through the cell membrane and can be utilized. For fermentation, the selection of membrane is itself a challenge as symmetric, porous, and asymmetric membranes. All of them have advantages and disadvantages like a system with higher permeability might prefer an asymmetric membrane but still, shape and configuration of the membrane must also check for mass transfer and efficient conversion [89]. Roy and colleagues modified the gas inlet system in a continuous stirred tank bioreactor system. The aeration tube was fitter at the bottom throughout the periphery of the reactor and have multiple discharge points. The effluent was taken off from the top of the reactor. The system work with mechanical stirring. Multi-point inject discharge adds more gas to the reactor and mechanical mixing provides lateral movement to bubbles which increased the pathway traveled by them. It increased the interaction of gas with liquid and cells and thus utilization and yield increased with the mixing rate [90]. 

R2Q4: Some sections, e.g. syngas production, only discuss specific setups and only a few possible, technical options. Please ensure, when you want to cover all aspects of syngas fermentation, to provide a comprehensive picture of all technical approaches (e.g. gasification technologies, other syngas sources as Co-SOEC,...). Furthermore, it is not comprehensive, why certain literature sources were discussed in the review, e.g. in Table 2, and how this selection represents an extensive overview of the existing research results.

R2A4: We have added new information and discussed recent literature in most of the sections. We have tried to summarize most of the recent literature in our manuscript we were able to find using online resources. In table 2 we have discussed only recently published articles mostly from 2020 onwards. We have discussed more newly published articles in table 2.

R2Q5: The language and form has to be improved at certain points to meet scientific standards and make sure, that the messages of the sentences are understandable. (no slashes and &, Eq 1, "wiz" in line 75, line 104-109, "too" line 521...). Figure 1 does not illustrate all possible pathways and gives only an overview of one specific setup, whereas its additional information value is not clear.

R2A5: We have made corrections as reviewers suggested and the language standard of the manuscript was improved with the help of a native English speaker. I think the reviewer is talking about figure 2, we have shown different pathways in Fig 2. We have improved this figure by including more pathways.

R2Q6: Please make sure, that all used literature is cited properly and all information from other sources are marked as citations, e.g. Fig 2, line 484-492,...

R2A6: We have cited all the references from where we have collected information. Figure 2 is not copied from other papers, we have designed it ourselves on the bases of facts and information collected from different articles.

R2Q7: A more elaborate discussion of the techno-economic results in the literature is necessary. When comparing different obtainable prices, it is important to state the relevant assumptions for the calculations (e.g. assumed plant size) to enable proper comparisons. I suggest adding an extensive table, comparing a broader range of literature results, and giving more details about the assumptions.

R2A7:  Thank you for the kind suggestion. All possible details have been added to the section. We have added some more examples there but sufficient literature related to the techno-economic analysis of syngas to bioalcohol is not available, as most of the studies reported at lab scale and no study related to techno-economic analysis has been performed. Very few papers are available for syngas fermentation and alcohol production, but as the reviewer suggested we tried to include more recent studies. Page 20; lines 621-648.

‘The performance of a hybrid plant, working for synthetic natural gas production from biomass as well as carbon capture and storage. The plant was utilizing 6.25 dt/h of virgin biomass and produced 1.32 t/h of synthetic natural gas. The energy efficiency of the plant was 51.2% and stored 2.97 t/h of CO2. Simulation-based analysis suggested that the synthetic gas process becomes economical if it achieved the target of 92.14 £/MWh and with an integrated approach for larger operations the price was reduced by about 15% [98]. A hybrid approach was adopted for ethanol production from bagasse via biomass gasification followed by syngas fermentation. Aspen plus and MATLAB simulated the utilization of biomass for ethanol yield of 283 L/per dry tonne of bagasse along with 43% energy efficiency. The simulation was targeted for multiobjective optimisation, to improve energy efficiency and reduced cost. Considering typical discounted cash flow analysis lead to a minimum ethanol selling price of 0.69 $/L [99]. Medeiros et al., have evaluated the performance of a bubble column reactor for syngas fermentation followed by anhydrous ethanol recovery using artificial neural networks and a multi-objective genetic algorithm. The major factors considered for simulation and optimization were the investment required, the energy efficiency of a process, minimum selling price, and most important bioreactor productivity that influence the whole economics of the process. With integrated production and recovery approach achieved a production rate of 124-133 MML/year. The calculated MSP was 0.707-0.713 $/L with high mass transfer. [100]. Different strategies have been evaluated for syngas fermentation and ethanol production along with methane from crude glycerol. The study group has been categorized into three: bioethanol production without CO2 capture, bioethanol production with CO2 capture, and bioethanol with CO2 capture followed by biomethanation of captured CO2. For fermentation, syngas was produced by biomass gasification. The working volume for glycerol utilization was 50,000 metric tons per year. The techno-economic feasibility suggested that groups have energy efficiency within the range of 30.2% to 35.1% among which group 3 has the highest i.e. 35.1%. MSPs for bioethanol were reduced to $1.32/L (group 1), $1.4 /L (group 2), and $0.31/L (group 3) [101].

R2Q8: The conclusion section has to be improved significantly. What is the conclusion of the extensive literature review? (see "USP")

R2A8: As per the suggestion the conclusion has been updated. Page 21; lines 670-684.

Syngas offers an opportunity for microbial conversion to value-added products via a greener process. Chemical processes are somehow efficient but operated at typically high temperatures and pressure. In addition, the chemical process also utilizes toxic solvents and costly catalysts. Microbial organisms also act as a catalyst for the sequestration of CO as a carbon source and synthesize various metabolites via microbial metabolism like ethanol, microbial biomass, biodiesel, higher alcohol, etc. In comparison to conventional fermentation with lignocellulosic biomass, syngas/gas fermentation offers an advantage as no biomass hydrolysis is needed for feed/hydrolysate preparation. Hence the feedstock becomes much cheaper for cost-effective product synthesis. In the view of the industrial aspect, the process operates at mild operational conditions and has a lower cost in comparison to conventional fermentation. In terms of the environment, the process offers a way to transform the waste gases from industrial processes and fuel burning along with revenue generation. It might contribute in making an industrial process eco-friendly and economic at large scale production.

Reviewer 3 Report

In the submitted manuscript (NUMBER: sustainability-2184819), investigated the recent trends in syngas fermentation for the production of ethanol and other energy-related products. Techno-economic aspects is also evaluated for industrial as well as social acceptance of the process. Biomass gasification produces syngas mainly comprised of CO and H2 along with H2S, CO2, N2, and tar compounds. Inorganic carbon present in syngas as CO and CO2 can be utilized for the production of several value-added chemicals including ethanol, higher alcohol, fuels, and hydrogen. However, chemical sequestration operates at a high temperature of 300-500 °C and pressure of 3-5 MPa in presence of heavy metal catalysts. However, scale-up and commercialization of technology have some obstacles like efficient mass transfer, microbial contamination, consistency in syngas composition and characteristics, and clean-up process. This review summarizes the recent advances in syngas production and utilization with special consideration to alcohols and energy-related products along with challenges for scale-up. Therefore, this is a meaningful work.

The manuscript could be accepted after substantial revise as follows.

(1)   In your manuscript, the use of punctuation was irregular. For instance, on page 3 line 93, “Fig.1:” and on page 6 line 238 “Table 1:”, you had better to revise “Fig.1.” and “Table 1.”. In addition, on the page 10 Table 2, the use of “°C” also was irregular, please check it carefully.

(2)   On Page 9 lines 266-267, “Anaerobic bacteria follow ‘Wood-Ljungdahl pathway’ for utilization of CO and CO2 sequestration.” Could other bacteria such as aerobic bacteria follow the “Wood-Ljungdahl pathway” for utilization of CO and CO2 sequestration?

(3)   On Page 12 lines 309-311, “In order to improve the gaseous fermentation and ethanol yield, a mixed gas stream containing CO, CO2, and H2 was aerated to culture continuously along with supplementation with rabbit faeces? More importantly, the faeces of other animals would replace the rabbit faeces?

(4)   On Page 12 lines 315-316, “Fernández-Blanco and colleagues combined syngas fermentation and chain elongation for the synthesis of bio-aclohol and medium chain length fatty acids. What are the medium chain length fatty acids? please give some examples.

(5)   The theme of the manuscript was “A mini-review on syngas fermentation to bio-alcohols: current status and challenges”, however, on page 5 line 205, Chemical-based methods for syngas conversion into valuable products, you had better to revise “Chemical-based methods for syngas conversion into bio-alcohols”, and remove the description of syngas conversion into other valuable products.

(6)   Please summarize the advantages and disadvantages as well as the differences between chemical-based methods and biological-based technologies for syngas conversion into bio-alcohols.

Author Response

Response to reviewer # 3

Reviewer#3

R3Q1: In your manuscript, the use of punctuation was irregular. For instance, on page 3 line 93, “Fig.1:” and on page 6 line 238 “Table 1:”, you had better to revise “Fig.1.” and “Table 1.”. In addition, on the page 10 Table 2, the use of “°C” also was irregular, please check it carefully.

R3A1: We appreciate reviewer's comments to improve the standard of our manuscript. We have made all the corrections as suggested by the reviewer. The punctuations have been updated and the whole manuscript has been checked for uniformity.

R3Q2: On Page 9 lines 266-267, “Anaerobic bacteria follow ‘Wood-Ljungdahl pathway’ for utilization of CO and CO2 sequestration.” Could other bacteria such as aerobic bacteria follow the “Wood-Ljungdahl pathway” for utilization of CO and CO2 sequestration?

R3A2: We appreciate the observation. The anaerobes use ‘Wood-Ljungdahl pathway’ but under aerobic conditions, microorganisms have another route which varies in carboxydotrophs and chemolithoautotrophs. The fig 2 has been updated and pathways are described in the text. (Page 10; line 297-302). ‘Under aerobic conditions also, some of the microorganisms like chemolithoautotrophs and aerobic carboxydotrophs possess the potential to utilize carbon monoxide as the sole source. For chemolithoautotrophs, CO acts as a sole source of carbon as well as electrons and in the presence of carbon monoxide dehydrogenase enzyme CO oxidized to CO2 [29]. While aerobic carboxydotrophs, CO is oxidized by utilizing molecular oxygen. From here carbon monoxide can be sequestered via the pentose phosphate pathway [28].  

R3Q3:  On Page 12 lines 309-311, “In order to improve the gaseous fermentation and ethanol yield, a mixed gas stream containing CO, CO2, and H2 was aerated to culture continuously along with supplementation with rabbit faeces? More importantly, the faeces of other animals would replace the rabbit faeces?

R3A3: A valid point suggested by the reviewer. The faecal content may be the source for organic waste as well as a unique group of microorganisms. For better understanding, the content has been updated and interpreted. (Page 13; line 380-385).

‘The main reason for using faecal matter is the enrichment of responsible microorganisms that contribute to syngas fermentation and product yield. The faeces from herbivores like rabbit, and horse faeces [80] is usually rich the homoacetogens, which are considered efficient microorganisms for syngas fermentation. The addition of faecal matter increased the population of homoacetogens and thus improved the syngas fermentation.’ 

R3Q4:   On Page 12 lines 315-316, “Fernández-Blanco and colleagues combined syngas fermentation and chain elongation for the synthesis of bio-alcohol and medium chain length fatty acids. What are the medium chain length fatty acids? Please give some examples.

R3A4:  Medium chain length fatty acids are referred as fatty acids with a chain length of 6-12 carbons. Caproic acid and butyric acid are some of examples of medium chain length fatty acids. (Page 13; Line 386-387)

R3Q5:   The theme of the manuscript was “A mini-review on syngas fermentation to bio-alcohols: current status and challenges”, however, on page 5 line 205, “Chemical-based methods for syngas conversion into valuable products”, you had better to revise “Chemical-based methods for syngas conversion into bio-alcohols”, and remove the description of syngas conversion into other valuable products.

R3A5: As per the suggestion the section heading has been updated. (page 6; line 218)

R3Q6:   Please summarize the advantages and disadvantages as well as the differences between chemical-based methods and biological-based technologies for syngas conversion into bio-alcohols.

R3A6: The advantages and disadvantages of both chemical and biological methods have been updated as for chemical-page 8; line 254-265 and for biological the whole section 3 depicted the challenges.

‘It is clear from the above examples that chemical catalysis offered very high selectivity and tenability for syngas conversion to alcohols and other products but like other processes, it also operates at a very high-temperature range from 350-600 °C and pressure. In addition, the cost of catalysts is also very high in the case of Rh, Pd, Pt, etc., which need regeneration after regular intervals. The process also employed solvents which may be hazardous. Overall technical feasibility, process cost, and environmental suitability make it more challenging to find some eco-friendly alternatives. In order to make the conversion process sustainable and cost-effective over chemical catalysis, biological sequestrations have been adopted. This approach emphasized on utilization of major components of syngas i.e. CO, H2, and in some cases COx and NOx as well in regular metabolic pathways of microorganisms via assimilation into various value-added chemicals and microbial biomass.’

Round 2

Reviewer 2 Report

The submitted manuscript has been significantly improved. Nevertheless, several points have not been adressed yet. These revisions have to be implemented before the manuscript should be considered for publication:

- Still, the presented review discusses several literature sources, but lacks novelty and additional information on the topic. In the introduction, the already existing REVIEWS have to be listed and it is necessary to illustrate, which additional insights are presented with the current manuscript. These insights have to be compiled in the main part and summarized in the conclusion section, providing novel information or perspectives on the topic rather than only summarizing a selection of other articles. 

- Furthermore, the review has to be structured more focussed (why is the section of alternative, catalytic syngas utilization options so long?) and it is not clearly stated, why some pathways are discussed and others are not mentioned (e.g. syngas production via alternative pathways, gasification technologies - e.g. not all gasification technologies need pulverized biomass...). The authors should decide, whether the reiview aims at giving a broad overview (then it is not covering all possible pathways and options), or it pursues the objective to discuss one or few aspects of syngas fermentation to alcohols in more detail. 

- Please make sure, that scientific writing standards are met. Don't use any slashes ("/") or "etc.", rewrite Eq. 1 properly, check consistency of units (°F line 183, °C) as well as notations (e.g. line 613) and make sure, that all abbrevations and terms are explained, when they are used for the first time (and only then, e.g. syngas line 234)

- CO capture in line 65?

Author Response

Response to reviewer # 2

R2Q1: Still, the presented review discusses several literature sources, but lacks novelty and additional information on the topic. In the introduction, the already existing REVIEWS have to be listed and it is necessary to illustrate, which additional insights are presented with the current manuscript. These insights have to be compiled in the main part and summarized in the conclusion section, providing novel information or perspectives on the topic rather than only summarizing a selection of other articles. 

R2A1: We appreciate reviewer’s comments to improve the standard of our manuscript. We have discussed some of the recently published articles and provided a figure 1 to show the contribution of different countries in this research area. We have added the information in the introduction part as. “According to the Scopus database, total 439 articles (353 research articles and 86 review articles) have been published since 2014 on syngas production and its valorization using chemical and fermentation methods. The United State alone has published 98 articles, followed by China (86), Germany (56), South Korea (46), Spain (37), and Netherland (34) (Fig. 1 Visualization network of countries involved in this area of research). A recently published article by Maki-Arvela discusses about the use of various catalysts for syngas conversion into aviation fuel [32]. In another article, Liu et al., has discussed different catalytic routes for ethanol production from syngas [33]. Very few articles have focused on the production of energy-related products from syngas via microbial fermentation. There is a review article focused on a broad aspect of biochemicals and biofuel production form syngas fermentation [30]. In the current review, recent trends in syngas fermentation for the production of ethanol and other energy-related products are summarized. Various parameters that affect the syngas fermentation process are discussed and tech-no-economic aspects are also evaluated for industrial as well as social acceptance of the process.

R2Q2: Furthermore, the review has to be structured more focussed (why is the section of alternative, catalytic syngas utilization options so long?) and it is not clearly stated, why some pathways are discussed and others are not mentioned (e.g. syngas production via alternative pathways, gasification technologies - e.g. not all gasification technologies need pulverized biomass...). The authors should decide, whether the reiview aims at giving a broad overview (then it is not covering all possible pathways and options), or it pursues the objective to discuss one or few aspects of syngas fermentation to alcohols in more detail. 

R2A2: We have added the alternative catalytic syngas utilization section to provide a brief idea to our readers who belong to biological background, that along with biological routes there are other chemical routes also available for syngas valorization. Methane or natural gas steam reforming are the other methods available for syngas production, but these methods are not economical and there is no report where these methods have been used for syngas production and produced syngas further used as a feedstock for microbial fermentation to produce biofuel. So, in our review article, we have covered only the biomass gasification method. We agree with the author's comment that not in all gasification methods pulverized biomass is used, but we have discussed a specific example from an article here and a reference is provided in support. We have made the statement more general “Before processing/gasification, biomass is dried and pulverized (if needed) for better results followed by gasification at a high temperature (800-1000 °C) and pressure of 1-20 bar. The aim of the review article is not broad, its totally focused on the use of syngas for bioalcohol production using the fermentation approach so most of our content is focused on this only. We have tried our best to cover all the recent literature available on this topic.

R2Q3: Please make sure, that scientific writing standards are met. Don't use any slashes ("/") or "etc.", rewrite Eq. 1 properly, check consistency of units (°F line 183, °C) as well as notations (e.g. line 613) and make sure, that all abbrevations and terms are explained, when they are used for the first time (and only then, e.g. syngas line 234). CO capture in line 65?

R2A3: Thank you for your suggestion. According to our opinion, there is no such rule that we can’t use /, etc. in our article. Where it's necessary these types of symbols can be used to make the information concise. We have rewritten the equation as

Biomass + O2 CO + H2 + CO2 + H2O + CH4

°F is changed to ºC to keep the uniformity in the text. We have also made other corrections as suggested by the reviewer.

Reviewer 3 Report

Accept!

Author Response

R3Q1: Accept

R3A1: We are thankful to the reviewer to evaluate our manuscript and recommend it for publication.
